# Acute Stroke Care in Mexico City: The Hospital Phase of a Stroke Surveillance Study

**DOI:** 10.3390/brainsci12070865

**Published:** 2022-06-30

**Authors:** Emmanuel Aguilar-Salas, Guadalupe Rodríguez-Aquino, Katya García-Domínguez, Catalina Garfias-Guzmán, Erika Hernández-Camarillo, Nayeli Oropeza-Bustos, Rubí Arguelles-Castro, Ameyalli Mitre-Salazar, Gloria García-Torres, Marco Reynoso-Marenco, Eduardo Morales-Andrade, Luis Gervacio-Blanco, Víctor García-López, Gabriel Valiente-Herves, Manuel Martínez-Marino, Fernando Flores-Silva, Erwin Chiquete, Carlos Cantú-Brito

**Affiliations:** 1Department of Neurology and Psychiatry, Instituto Nacional de Ciencias Médicas y Nutrición Salvador Zubirán, Mexico City 14080, Mexico; emmanuel_aguilarsls@hotmail.com (E.A.-S.); donajyrdz@gmail.com (G.R.-A.); garcia_katya@outlook.com (K.G.-D.); ihati.uva@gmail.com (C.G.-G.); ealejandra2308@gmail.com (E.H.-C.); ileyanbella@gmail.com (N.O.-B.); rubi.ac894@gmail.com (R.A.-C.); atzinsalas@gmail.com (A.M.-S.); ferfs98@hotmail.com (F.F.-S.); 2Department of Neurology, Hospital de Especialidades “Dr. Belisario Domínguez”, Mexico City 09930, Mexico; jensenanzahebd@gmail.com; 3Department of Internal Medicine, Hospital General “Dr. Darío Fernández Fierro”, Instituto de Seguridad y Servicios Sociales de los Trabajadores del Estado, Mexico City 03900, Mexico; mreynosomarenco@yahoo.com; 4Department of Epidemiology, Hospital General “Dr. Darío Fernández Fierro”, Instituto de Seguridad y Servicios Sociales de los Trabajadores del Estado, Mexico City 03900, Mexico; andradre.ema@hotmail.com; 5Department of Emergencies, Hospital General “Dr. Darío Fernández Fierro”, Instituto de Seguridad y Servicios Sociales de los Trabajadores del Estado, Mexico City 03900, Mexico; mcdrger15@gmail.com; 6Department of Internal Medicine, Hospital General Tlahuac, Mexico City 13250, Mexico; victorregal@yahoo.com.mx; 7Department of Internal Medicine, Hospital General de Zona 47, Instituto Mexicano del Seguro Social, Mexico City 09200, Mexico; gabriel106@hotmail.com; 8Department of Neurology, Hospital General de Zona 32, Instituto Mexicano del Seguro Social, Mexico City 04980, Mexico; manuelmtzm@hotmail.com

**Keywords:** acute stroke care, stroke, surveillance, hospital, Mexico

## Abstract

Background: Acute stroke care has greatly improved in recent decades. However, the increasing stroke mortality in low-to-middle income countries suggests that progress has not been reached completely by these populations. Here we present the analysis of the hospital phase of the first population-based stroke surveillance study. Methods: A daily hospital surveillance method was used to identify adult patients with acute stroke during 18 months in six hospitals. We abstracted data on demographics, vascular risk factors, neuroimaging-confirmed stroke types, and clinical data. Results: A total of 1361 adults with acute stroke were identified (mean age 69.2 years; 52% women) with transient ischemic attack (5.5%), acute ischemic stroke (68.6%), intracerebral hemorrhage (23.1%), cerebral venous thrombosis (0.2%), and undetermined stroke (2.6%). The main risk factors were hypertension (80.7%) and diabetes mellitus (47.6%). The usage rate of thrombolysis was 3.6%, in spite of the fact that 37.2% of acute ischemic stroke patients arrived in <4.5 h. The 30-day case fatality rate was 32.6%, higher in hemorrhagic than ischemic stroke. Conclusion: We identified limitations in acute stroke care in the Mexico City, including neuroimaging availability and thrombolysis usage. The door-to-door phase will help to depict the acute stroke burden in Mexico.

## 1. Introduction

Stroke remains the second leading cause of general mortality and the third leading cause of disability worldwide [1]. Although progress has been documented in high-income countries during the past several decades, incidence and mortality of stroke has been increasing in low-to-middle income countries [1].

Different interventions and strategies have been proposed to improve the knowledge, prevention, and treatment of cerebrovascular diseases. One of these strategies is the implementation of population-based studies, hospitalized stroke surveillances, and multicenter registries that provide valuable information to analyze the frequency, risk factors, etiologies, acute care, hospital management, and clinical outcomes that might help to construct healthcare policies and clinical guidelines [2,3]. To carry out a population-based study in large urban metropolises, it is necessary to cover all the hospitals that serve this population to capture all the cases of those who seek hospital care. To depict comprehensively the stroke burden, it is not only necessary to account for the number of cases, but also to describe disabilities, case-fatality rate (CFR) and population mortality, therapeutic strategies, and characteristics of timely acute stroke care [3].

In Mexico, the only acute stroke population-based study was performed more than 10 years ago (i.e., the Brain Attack Surveillance in Durango, BASID study). The BASID study showed that cumulative incidence was 232.2 per 100,000 persons—years among people 35 years and older, with a prevalence of 8 per 1000 [4,5]. Some hospital registries have also been performed in different years and regions of Mexico describing the frequency of stroke types, associated risk factors, and some characteristics of hospital care [6,7,8,9].

The World Health Organization (WHO) suggests that population-based studies and hospital registries should be carried out periodically in order to describe the trends in the standards of stroke care in a particular country [10]. The aim of this study is to describe the hospital phase (first phase) of the comprehensive Brain Attack Surveillance in Tláhuac municipality (BASIT) study, the first population-based study of Mexico City. The second phase has already been completed and is under analysis, which will provide the population incidence, prevalence, and the population mortality of the different acute stroke subtypes. The present hospital phase is aimed to describe characteristics of hospital care, resource usage, patients’ flow and disposition, mid-term outcome, and the distribution of hospitalization rates among the stroke subtypes.

## 2. Materials and Methods

### 2.1. Study Population

A daily active capture of adult patients with acute stroke was carried out during 18 consecutive months, as the first phase of the population-based BASIT study. This multicenter active data capture project was arranged according to current WHO standards and recommendations [10]. Tláhuac is one of the 16 districts or municipalities of Mexico City. Tláhuac was selected because its population is well contained within the large metropolitan area and hospital care is provided in specific healthcare districts on a captive population. For the hospital phase, we selected the six hospitals that provide health care to this administratively captive community. These hospitals also provide medical care to other adjacent Mexico City municipalities (Figure 1). In order to ensure the most complete identification of stroke patients, both active (i.e., “hot pursuit”) and passive (i.e., “cold pursuit”) surveillance was applied on a daily basis at all participating hospitals. For active surveillance, emergency rooms (ERs), admission logs, hospital wards, and intensive care units (ICUs) were reviewed every day by study medical staff. This staff was comprised of general practitioners who received standardized training on stroke including certification in NIHSS and the modified Rankin scale (mRS) before starting the study. For passive surveillance, lists of discharges with International Classification of Diseases, Tenth Revision (ICD-10) codes I60 to I69 (i.e., cerebrovascular disease) were obtained monthly from each hospital. Any newly identified stroke patients were reviewed and included in the database. The Steering Committee BASIT neurologists validated cases with inconsistent or conflicting stroke data. Exclusion criteria were subjects younger than 18 years or individuals who had head trauma. Those patients residing in the Tláhuac municipality are included in the second phase BASIT study.

### 2.2. Ethics and Oversight

The BASIT study was approved by the following committees: Research and Ethics Committee of the Instituto Nacional de Ciencias Médicas y Nutrición Salvador Zubirán, the Instituto Mexicano del Seguro Social, the committee of the Instituto de Seguridad y Servicios Sociales de los Trabajadores del Estado and the Secretaría de Salud. The patients or their legal proxies signed informed consent and all of the investigators pertaining to the BASIT staff were required special permission to abstract study data on site.

### 2.3. Data Collection

The study medical staff used a computerized registry instrument where all data were entered directly into an on-site laptop-based database that included variables such as demographic data (sex, age, and place of residence), vital signs upon admission (blood pressure, capillary glucose, and body temperature) vascular risk factors, previous medications, stroke subtypes based on neuroimaging findings, laboratory studies performed during the hospital stay, clinical severity, the probable mechanisms and causes for each stroke subtype, hospitalization length, and in-hospital complications (i.e., infections, seizures, mental status impairment, and the need for invasive mechanical ventilation (IMV)).

We used the National Institutes of Health Stroke Scale (NIHSS) to assess the severity of patients with acute ischemic stroke (AIS). Patients with intracerebral hemorrhage (ICH) and subarachnoid hemorrhage (SAH) were evaluated with the Glasgow coma scale (GCS). In patients with SAH, their clinical status was stratified by using the Hunt–Hess scale, and by computed tomography (CT) scan was classified with the Fisher scale, mainly for prediction of cerebral vasospasm. In patients with transient ischemic attack (TIA), the ABCD^2^ scale was obtained to predict the risk of AIS within the following days. All the clinical scales used in the study were registered in the clinical records in real time and further validated with subsequent daily visits by the research staff. The functional outcome was assessed by using the mRS at 30 days, considered since the date of the symptoms’ onset. The clinical assessments with the mRS were performed physically if the patient remained at the hospital by day 30, or by telephone if already discharged. In the cases of fatal stroke, the cases were registered on a daily basis and confirmed with the death records of each hospital.

### 2.4. Definitions

Stroke was defined as an acute and sudden neurological deficit attributed to a CNS focal due to vascular occlusion or rupture and confirmed with neuroimaging. Strokes were classified into TIA, AIS, ICH, SAH, or cerebral venous thrombosis (CVT) according to neuroimaging findings. TIA was defined as a focal CNS neurovascular syndrome, which resolves completely in less than 24 h, with no other apparent cause than the intracerebral artery occlusion and neuroimaging study ruling out other etiologies or other stroke subtypes. Undetermined stroke (UDS) was defined as an acute episode of neurological dysfunction that was suspected to be of ischemic or hemorrhagic origin and that persisted for more than 24 h or until its death, but the neuroimaging study could not be performed [11]. Patients who woke up with a neurovascular syndrome were defined as wake-up strokes, and their onset time was considered as the last time that the patient was seen well.

### 2.5. Statistical Analysis

Descriptive analyses of the qualitative variables were expressed as proportions. Quantitative variables are presented as arithmetic means with standard deviations (SD), or medians with interquartile ranges (IQR), depending on whether they presented a normal distribution or not. For the comparative analysis on each stroke subtype, the chi-square test was used for qualitative variables and the ANOVA or Kruskal–Wallis test was used for quantitative variables in more than two groups. Statistical significance was assumed with a level of significance at *p* < 0.05. Unadjusted and adjusted hazard ratios with their respective 95% confidence intervals (CIs) were calculated to assess the relationship of relevant risk factors with outcomes of interest. All data analysis was performed by using SPSS version 23 (IBM Corp. in Armonk, NY, USA).

## 3. Results

From 1 April 2018 to 30 September 2019, a total of 1361 patients (52.1% women, mean age: 69.2 ± 14.3 years) with acute stroke were included: 75 (5.5%) cases with TIA, 934 (68.6%) with AIS, 251 (18.4%) ICH, 63 (4.6%) SAH, 3 (0.2%) CVT, and 35 (2.6%) with UDS. Wake-up stroke occurred in 112 (8.2%) patients. There were 208 (15.3%) patients with a previous stroke: 174 (12.8%) AIS and 34 (2.5%) ICH cases. In all, 84 (6.2%) cases occurred in young (≤45 years old) and 336 (24.7%) in very elderly (≥80 years old) patients. Women were older than men (mean age 70.6 vs. 67.6 years in women and men, respectively; *p* < 0.001) (Figure 2A). The frequency of stroke subtypes varied among age groups, mainly because hemorrhagic strokes were more common in younger patients (*p* < 0.001) (Figure 2B).

Table 1 describes the demographic data, vascular risk factors, and complications by stroke subtype (the three patients with CVT were excluded from this analysis). The most prevalent vascular risk factor was systemic arterial hypertension, reaching up to 80%. In addition, diabetes mellitus was present in 50% of patients with AIS. Dyslipidemia was found in 68% among 454 (33.4%) patients with plasma lipids measured or who were under medication for hypercholesterolemia or hypertriglyceridemia. In patients with AIS, the most prevalent heart risk factors were atrial fibrillation (11.3%), ischemic heart disease (7.7%) and chronic heart failure (2.6%). Chronic kidney disease was observed in 11% of patients with ICH and 9% in AIS. With regard to hospital stay, most (72%) of the patients with TIA were discharged from the ER. In-hospital infections (pneumonia, urinary tract infections, and sepsis) were identified in around 25% of patients, depending on the stroke subtype. IMV was required in 262 (19.3%) patients, but 190 (72.5%) received IMV at the internal medicine wards, and 70 (26.7%) at the ER, whereas only 22 (8.4%) were admitted to the ICU.

Patients with AIS had a median NIHSS scoring of 12 points (IQR: 7–20 points), but only 34 out of 934 patients (3.6%) received intravenous thrombolysis with tPA in spite of the fact that 347 (37.2%) arrived within 4.5 h. No patients underwent mechanical thrombectomy. Diagnostic evaluation of patients with AIS included a CT scan in 919 patients (98.4%), MRI in 25 (2.7%), electrocardiogram in 635 (68%), echocardiogram in 84 (9%), 24-h Holter monitoring in 22 (2.4%), and carotid Doppler ultrasound in 46 (4.9%). Because of the low usage rate of diagnostic studies, the etiology of AIS was determined in only 218 (23.3%) cases (Table 2). The most frequently involved vascular territories were the middle cerebral artery in 754 (80.7%) patients, posterior cerebral artery in 96 (10.3%), any cerebellar artery in 51 (5.5%), and basilar artery in 22 (2.4%) AIS cases.

Patients with ICH had a median GCS score of 10 points (IQR: 6–15). ICH had a lobar location in 111 (41.3%) patients, basal ganglia in 79 (29.4%), thalamus in 38 (14.1%), cerebellum in 17 (6.7%), brainstem in 15 (5.6%), and 8 (3%) were primary intraventricular hemorrhages. Ventricular extension occurred in 101 patients (40.2%). The main cause of ICH was related to uncontrolled arterial hypertension (84%), including 15.5% of patients who did not know their hypertension diagnosis. Other ICH etiologies are described in Table 2.

Patients with SAH had a median GCS score of 7 points (IQR: 5–12). In all, 39.7% patients with SAH had a Hunt–Hess class V, and 73% were classified with a Fisher scale grade IV. A cerebral aneurysm could be documented in only 23 patients (36.5%) due to insufficient use of cerebral angiography. The most common location of aneurysms was the bifurcation of the middle cerebral artery (*n* = 10), posterior communicating artery (*n* = 6), anterior communicating artery (*n* = 5), and basilar artery (*n* = 2).

Figure 3 shows the clinical outcome according to stroke subtype. CFR in patients with AIS was 28.7% (95% CI: 25.8–31.6), in ICH 50.2% (95% CI: 44.0–56.4), in SAH 57.1% (95% CI: 44.9–69.3), and in the UDS subtype 34.3% (95% CI: 18.9–49.5). The CFR for CVT was 33.3% (95% CI: 20.0–86.0). The cause of death in patients with TIA was non neurological in all cases. Patients with chronic heart failure (OR 1.57, 95% CI: 1.11–2.22, *p* = 0.026) and with chronic kidney disease (OR 1.6, 95% CI: 1.31–1.95, *p* =< 0.001) were significantly associated with death after 30 days of follow-up. Significant disability (mRS 3 to 5) in patients with AIS was 58.1% (95% CI: 54.8–61.1), hemorrhagic stroke 39.5% (95% CI: 33.6–44.3), and UDS type 45.7% (95% CI: 28.5–61.4).

## 4. Discussion

In this first stroke surveillance study carried out in general hospitals of the Mexico City, several major points emerge regarding deficiencies in acute stroke care that is associated with a high short-term CFR, a characteristic previously identified [4]. These findings reflect the critical public health challenges that need to be met to provide access to acute and long-term stroke care for a large proportion of the Mexican population. Although several healthcare centers might have made improvements, Mexico still lacks standardized management in the different phases of stroke care to cover the entire population, including neuroimaging availability, stroke codes, implementation of stroke units, and thrombolysis usage. Indeed, the rate of intravenous thrombolysis in patients with acute cerebral infarction remains too low, with a slight increase from 1.1% to 3.6% during the last several decades [6,9]. This current rate is considerably lower than that reported in other countries [12,13]. Moreover, our low intravenous thrombolysis rate occurred in spite of the fact that around 37.7% of patients arrived at the ER within 4.5 h of stroke onset. One explanation for insufficient use of thrombolysis was the lack of 24/7 neuroimaging availability. In addition, no patient underwent mechanical thrombectomy because this therapeutic option was not already implemented at the participating centers. The lack of acute stroke care systematization and implementation of the current evidence-based acute therapeutic strategies unveils the limitations of a fragmented healthcare system (in Mexico, there are seven central autonomous public healthcare institutions) not completely covering the entire population. Moreover, there are notable differences in the celerity and quality of care in public vs. private hospitals and emergency care providers in Mexico [14,15,16].

Arterial hypertension and diabetes mellitus are the main stroke risk factors, with a notable increase in diabetes mellitus frequency up to 50% in AIS patients, compared with the reported 35–40% in previous Mexican stroke registries [6,9,16]. These data are in accordance with the high prevalence of these risk factors in Mexico. From 2012 to 2018 the prevalence of arterial hypertension increased from 16.6% to 18.4% and diabetes mellitus from 9.2% to 10.3%, respectively [17]. This growth is due to the increase in obesity, high carbohydrate diets, and scarce physical activity [17]. Therefore, these two chronic diseases are still the main stroke risk factors, which is consistent with a previous population study conducted in 2008. The high prevalence of hypertension and insufficient control explain the high proportion of hemorrhagic strokes, particularly in younger patients, compared with populations in Europe [18,19,20], and Asia [21,22], but with notable similarities when compared with other Latino and Japanese populations [23,24,25,26].

In contrast, current smoking was not a relevant risk factor for stroke in our study, which is in agreement with the low smoking prevalence in our population (28.4% in men and 9.2% in women) [17], compared with the European population (43.5% in men and 23.4% in women) [27], but comparable with the American population (25.8% in men and 14.1% in women) [28]. Also, given that alcoholic liver disease is the main cause of death in Mexicans pertaining to the age group 35–55 years, alcoholism had a low prevalence in our study, possibly due to the older age for stroke presentation [29].

Another important observation in the present study is that most (76.7%) AIS cases were classified as of undetermined etiology, due to limited use of diagnostic resources that are of paramount importance to guide secondary stroke prevention. This is evidenced by the negligible (<2%) frequency of the large artery disease mechanism, since Doppler ultrasound was performed in <5% of the patients.

Finally, we found a high (50.5%) frequency of moderate-to-severe disability 30 days after stroke, as compared with other stroke registries [25]. Several facts may explain this observation, including the lack of early in-hospital rehabilitation programs to help patients in attaining the maximum possible recovery. Also, many patients are discharged to home from the ER (70% in AIS), losing the opportunity to receive a stroke care, with early in-hospital rehabilitation, dysphagia testing, and diagnostic studies to assign AIS etiology. The main challenge in implementing this population-based hospital surveillance phase study was to coordinate all the hospitals that belong to different health systems in Mexico and to guarantee the correct registration of patients with stroke and to exclude those who had a stroke mimic.

The main limitation of the present study is that we present information on the first phase of this population-based project based on hospitalized stroke patients. In the second door-to-door surveillance phase, we will also present information on patients not reaching the hospital system. It is possible that, on one hand, some patients with mild strokes received medical care in outpatient clinics, and on the other hand, some severe strokes caused death before receiving medical care at a referral center; the latter is especially possible in hemorrhagic strokes (i.e., SAH or ICH) [30]. In the door-to-door stroke survey from the antecedent BASID study in Durango, Mexico, around 23% of the patients with acute stroke were not hospitalized. Another limitation of this study is the relatively low number of cases, particularly for the less frequent stroke subtypes, such as CVT cases, which hampers detailed studies on the characteristics of hospital care of these cases.

## 5. Conclusions

The current study presented a multicenter hospital stroke registry in the city with the largest population in Mexico that is derived from a population-based surveillance study of stroke. Compared to a study carried out 12 years ago in the north of our country, stroke has not decreased in its frequency, mortality has remained the same, and treatment strategies have not increased significantly. Several uncovered needs in acute stroke care may explain the lack of improvement during the last decade in Mexico.

## Figures and Tables

**Figure 1 brainsci-12-00865-f001:**
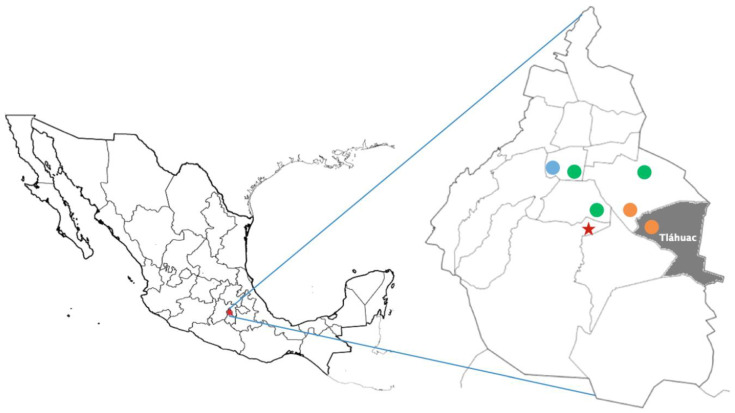
Map of Mexico (**left**), and a map of Mexico City (**right**). The BASIT study was performed in Tláhuac municipality (gray zone), which is located in the Mexico City. Orange, green, and blue circles represent the hospitals from the Secretaría de Salud, the Instituto Mexicano del Seguro Social, and the Instituto de Seguridad y Servicios Sociales de los Trabajadores del Estado, respectively. Red star: the location of the Instituto Nacional de Ciencias Médicas y Nutrición Salvador Zubirán.

**Figure 2 brainsci-12-00865-f002:**
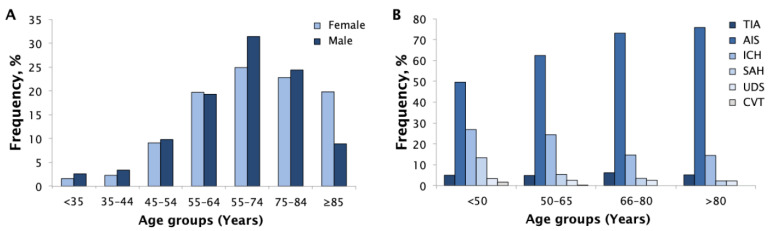
Proportions by age group and sex (**A**) and proportions by age groups and stroke subtypes (**B**). AIS, acute ischemic stroke; CVT, cerebral venous thrombosis; ICH, intracerebral hemorrhage; SAH, subarachnoid hemorrhage; TIA, transient ischemic attack; UDS, undetermined stroke.

**Figure 3 brainsci-12-00865-f003:**
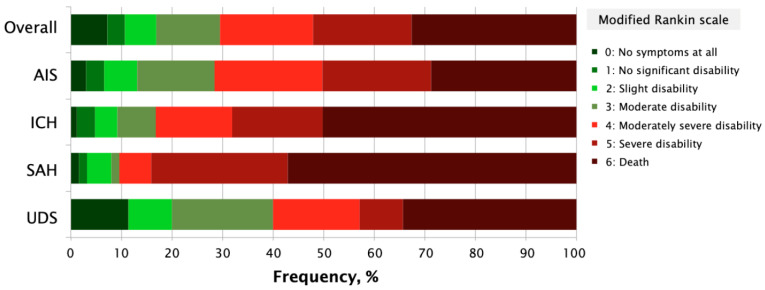
Functional outcome at 30-day follow-up by stroke subtype. ICH indicates intracerebral hemorrhage; AIS, acute ischemic stroke; mRS, modified Rankin scale; SAH, subarachnoid hemorrhage; and UDS, undetermined stroke.

**Table 1 brainsci-12-00865-t001:** Baseline clinical and demographic characteristics and in acute stroke patients.

	Total	TIA	AIS	ICH	SAH	UDS	*p* Value
	(*n* = 1358)	(*n* = 75)	(*n* = 934)	(*n* = 251)	(*n* = 63)	(*n* = 35)	
Age, median (IQR), years	70 (60–79)	70 (61–80)	72 (63–81)	65 (56–77)	62 (47–70)	70 (62–78)	<0.001
Sex at birth, *n* (%)							0.147
Women	707 (52.1)	46 (61.3)	479 (51.3)	126 (50.2)	40 (63.5)	16 (45.7)	
Men	651 (47.9)	29 (38.7)	455 (48.7)	125 (49.8)	23 (36.5)	19 (54.3)	
Hospital stay, median (IQR), days	7 (3–13)	2 (1–4)	7 (4–13)	8 (4–17)	9 (4–38)	4 (2–7)	<0.001
Wake-up stroke, *n* (%)	112 (8.2)	3 (4.0)	89 (9.7)	11 (4.4)	6 (9.5)	3 (8.8)	0.057
Vascular risk factors, *n* (%)							
Arterial hypertension	1099 (80.7)	61 (81.3)	760 (81.4)	208 (82.9)	39 (61.9)	31 (88.6)	<0.001
Diabetes mellitus	647 (47.6)	36 (48.0)	477 (51.1)	99 (39.4)	17 (27.0)	18 (51.4)	<0.001
Dyslipidemia *	289/454 (63.7)	20/24 (83.3)	213/334 (63.8)	45/81 (55.6)	7/10 (70.0)	3/4 (75.0)	0.200
Current smoking	219 (16.1)	8 (10.7)	156 (16.7)	44 (17.5)	9 (14.3)	2 (5.7)	0.367
Chronic alcoholism	146 (10.8)	6 (8.0)	101 (10.8)	28 (11.2)	7 (11.1)	4 (11.4)	0.929
Non-valvular atrial fibrillation	118 (8.7)	3 (4.0)	106 (11.3)	7 (2.8)	1 (1.6)	1 (2.9)	<0.001
Ischemic heart disease	93 (6.8)	7 (9.3)	72 (7.7)	12 (4.8)	0	2 (5.7)	0.082
Chronic heart failure	35 (2.6)	4 (5.3)	24 (2.6)	3 (1.2)	3 (4.8)	1 (2.9)	0.250
Peripheral vascular disease	27 (2.0)	0	23 (2.5)	2 (0.8)	1 (1.6)	1 (2.9)	0.176
Chronic kidney disease	124 (9.1)	7 (9.3)	86 (9.2)	27 (10.8)	2 (3.2)	2 (5.7)	0.389
Prior stroke	208 (15.3)	19 (25.3)	145 (15.5)	23 (9.2)	2 (3.2)	5 (14.3)	<0.001
Hospital care services, *n* (%)							
Only emergency room	478 (35.0)	54 (72.0)	319 (33.9)	65 (25.5)	9 (14.3)	31 (88.6)	<0.001
Internal medicine service	853 (62.8)	21 (28.0)	606 (64.9)	177 (70.5)	45 (71.4)	4 (11.4)	<0.001
Intensive care unit	27 (2.0)	0	9 (1.0)	9 (3.6)	9 (14.3)	0	<0.001
In-hospital complications, *n* (%)							
Pneumonia	144 (10.6)	1 (1.3)	83 (8.9)	39 (15.5)	20 (31.7)	1 (2.9)	<0.001
Urinary tract infections	173 (12.7)	2 (2.7)	127 (13.6)	24 (9.6)	13 (20.6)	7 (20)	0.006
Sepsis	68 (5.0)	1 (1.3)	49 (5.2)	13 (5.2)	4 (6.3)	1 (2.9)	0.573
Seizures	27 (2.0)	0	18 (1.9)	5 (2.0)	4 (6.3)	0	0.088
Altered mental status	381 (28.1)	2 (2.6)	188 (20.1)	131 (52.2)	52 (82.5)	8 (22.9)	<0.001
Invasive mechanical ventilation	262 (19.3)	1 (1.3)	105 (11.2)	104 (41.4)	47 (74.6)	5 (11.4)	<0.001

* Figure numbers among total tested for plasma lipids. AIS, acute ischemic stroke; ICH, intracerebral hemorrhage; IQR, interquartile range; LHS, length of hospital stay; SAH, subarachnoid hemorrhage; TIA, transient ischemic attack; UDS, undetermined stroke.

**Table 2 brainsci-12-00865-t002:** Stroke subtypes and etiologies.

Stroke Subtype and Etiological Category, *n* (%)	*n* = 1358
Ischemic stroke	*n* = 934
Cardioembolism	184 (19.7)
Large-artery atherosclerosis	17 (1.8)
Other determined etiology	16 (1.7)
Tumors	7 (0.8)
Hypercoagulability state	6 (0.6)
Arterial dissection	2 (0.2)
Vasculitis	1 (0.1)
Small-vessel occlusion	1 (0.1)
Undetermined cause	714 (76.7)
Incomplete workshop	710 (76)
Cryptogenic	4 (0.7)
Intracerebral hemorrhage	*n* = 251
Arterial hypertension	211 (84)
Unknown	21 (8.4)
Tumors	10 (4.0)
Anticoagulants	5 (2.0)
Arteriovenous malformation	3 (1.2)
Hematologic disease	1 (0.4)
Subarachnoid hemorrhage	*n* = 63
Unknown	36 (57.1)
Aneurysm	23 (36.5)
Arteriovenous malformation	2 (3.2)
Anticoagulants	1 (1.6)
Perimesencephalic hemorrhage	1 (1.6)
Cerebral venous thrombosis	*n* = 3
Cancer	1 (33.3)
Hypercoagulability state	1 (33.3)
Pregnancy	1 (33.3)

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
