# Peer review of "Acute Stroke Care in Mexico City: The Hospital Phase of a Stroke Surveillance Study"

_brainsci, 2022, doi:10.3390/brainsci12070865_

Round 1

Reviewer 1 Report

I think the study is very important, devastating numbers to a stroke neurologist in Europe, and by publishing it hopefully it can raise political awareness and induce change in Stroke care in Mexico city. 

Author Response

The authors profoundly thank the reviewer's suggestions for our paper. We will make an effort to improve the original version of the paper by revising the suggestions and making these and further changes accordingly. Below are the responses to the reviewer's comments.

Reviewer 1

I think the study is very important, devastating numbers to a stroke neurologist in Europe, and by publishing it hopefully it can raise political awareness and induce change in Stroke care in Mexico city. 

Response: Thank you very much for your comments.

Reviewer 2 Report

Overall interesting and well written study regarding Acute Stroke Care in Mexico City. 

The abstract highlight each area of the manuscript and it contains a defined purpose and a consistent conclusion.    The introduction gives adequate background information and a well-defined hypothesis or purpose.    Clinical, experimental, and statistical methods are adequate to address the stated hypothesis.    Results are presented in a logical fashion and seem valid.   The discussion presents results in the wider context of what has been investigated in this area before.    Figures and tables adequately demonstrate and display the findings.   The conclusion succinctly highlights the major points of the paper.   References are up-to-date, complete, and properly cited.

Author Response

The authors profoundly thank the reviewer's suggestions for our paper. We will make an effort to improve the original version of the paper by revising the suggestions and making these and further changes accordingly. Below are the responses to the reviewer's comments.

Reviewer 2

Overall interesting and well written study regarding Acute Stroke Care in Mexico City. 

The abstract highlight each area of the manuscript and it contains a defined purpose and a consistent conclusion.    The introduction gives adequate background information and a well-defined hypothesis or purpose.    Clinical, experimental, and statistical methods are adequate to address the stated hypothesis.    Results are presented in a logical fashion and seem valid.   The discussion presents results in the wider context of what has been investigated in this area before.    Figures and tables adequately demonstrate and display the findings.   The conclusion succinctly highlights the major points of the paper.   References are up-to-date, complete, and properly cited. 

Response: Thank you very much for your comments. We will make an effort to double-check the manuscript and improve it further.

Reviewer 3 Report

This article reveals the reality of stroke treatment in developing countries.

Methods are appropriately described and there are no problems with statistical methods.

However, I have some suggestions to improve this paper.

In the discussion, the authors cite the lack of medical resources in Mexico as the reason for the differences from stroke databases in developed countries.

Is this the only reason? Shouldn't the authors also explain the Mexican insurance system and address health care system issues such as private ambulances?

An explanation of the economic situation in the region where the study was conducted might also help readers understand.

Minor revision

Line118 Rankin scale -> modified rankin scale

Table1 IS->AIS, UD ->UDS

Table1 I don't know what the "/" in the Dyslipidemia section means

It is difficult to read table2 because it is divided into two pages.

Line282-4 Please provide references on which the values are based.

Author Response

The authors profoundly thank the reviewer's suggestions for our paper. We will make an effort to improve the original version of the paper by revising the suggestions and making these and further changes accordingly.

Below are the responses to the reviewer's comments. The new version of our paper contains the changes highlighted in yellow.

Reviewer 3

This article reveals the reality of stroke treatment in developing countries.

Methods are appropriately described and there are no problems with statistical methods.

However, I have some suggestions to improve this paper.

In the discussion, the authors cite the lack of medical resources in Mexico as the reason for the differences from stroke databases in developed countries.

Is this the only reason? Shouldn't the authors also explain the Mexican insurance system and address health care system issues such as private ambulances?

Response: Thank you very much for this comment. Indeed, there are other possible causes that explain these findings, and among them, are those that Reviewer 3 is citing. In the new version of the manuscript, we include a brief discussion on this topic.

An explanation of the economic situation in the region where the study was conducted might also help readers understand.

Response: We completely agree and have made this change accordingly.

Minor revision

Line118 Rankin scale -> modified rankin scale

Response: We have corrected this in the new version of the paper.

Table1 IS->AIS, UD ->UDS

Response: We have corrected this in the new version of the paper.

Table1 I don't know what the "/" in the Dyslipidemia section means

Response: We indicated the percent among the number of patients tested (since not all had tested for this biomarker). For example, we used "289/454 (63.7)" to express that 289 patients among 454 tested (i.e., 63.7%) had dyslipidemia. We have added a footnote to explain this in the new version of the paper.

It is difficult to read table2 because it is divided into two pages.

Response: We agree. We will resize Figure 1 in order to Table 2 is not divided by the page change.

Line282-4 Please provide references on which the values are based.

Response: Thank you. We have now added the proper references.